# Impact of Yeast-Derived β-Glucans on the Porcine Gut Microbiota and Immune System in Early Life

**DOI:** 10.3390/microorganisms8101573

**Published:** 2020-10-13

**Authors:** Hugo de Vries, Mirelle Geervliet, Christine A. Jansen, Victor P. M. G. Rutten, Hubèrt van Hees, Natalie Groothuis, Jerry M. Wells, Huub F. J. Savelkoul, Edwin Tijhaar, Hauke Smidt

**Affiliations:** 1Laboratory of Microbiology, Wageningen University, 6700 EH Wageningen, The Netherlands; hugo.devries@wur.nl; 2Host-Microbe Interactomics Group, Wageningen University, 6700 AH Wageningen, The Netherlands; jerry.wells@wur.nl; 3Cell Biology and Immunology Group, Wageningen University, 6700 AH Wageningen, The Netherlands; mirelle.geervliet@wur.nl (M.G.); natalie.groothuis@hotmail.com (N.G.); huub.savelkoul@wur.nl (H.F.J.S.); edwin.tijhaar@wur.nl (E.T.); 4Department of Biomolecular Health Sciences, Division of Infectious Diseases and Immunology, Faculty of Veterinary Medicine, Utrecht University, 3584 CL Utrecht, The Netherlands; c.a.jansen@uu.nl (C.A.J.); v.rutten@uu.nl (V.P.M.G.R.); 5Department of Veterinary Tropical Diseases, Faculty of Veterinary Science, University of Pretoria, Private Bag X04, Onderstepoort 0110, South Africa; 6Research and Development, Trouw Nutrition, 3800 AG Amersfoort, The Netherlands; hubert.van.hees@trouwnutrition.com

**Keywords:** β-glucans, porcine, gastro-intestinal tract, gut microbiota, immune system, early life

## Abstract

Piglets are susceptible to infections in early life and around weaning due to rapid environmental and dietary changes. A compelling target to improve pig health in early life is diet, as it constitutes a pivotal determinant of gut microbial colonization and maturation of the host’s immune system. In the present study, we investigated how supplementation of yeast-derived β-glucans affects the gut microbiota and immune function pre- and post-weaning, and how these complex systems develop over time. From day two after birth until two weeks after weaning, piglets received yeast-derived β-glucans or a control treatment orally and were subsequently vaccinated against *Salmonella* Typhimurium. Faeces, digesta, blood, and tissue samples were collected to study gut microbiota composition and immune function. Overall, yeast-derived β-glucans did not affect the vaccination response, and only modest effects on faecal microbiota composition and immune parameters were observed, primarily before weaning. This study demonstrates that the pre-weaning period offers a ‘window of opportunity’ to alter the gut microbiota and immune system through diet. However, the observed changes were modest, and any long-lasting effects of yeast-derived β-glucans remain to be elucidated.

## 1. Introduction

Early in life and during weaning, pigs are at risk of developing diseases which compromise animal welfare and have major economic consequences for the pig industry. The early postnatal and weaning periods are critical, as the gastrointestinal (GI) system and the immune system are still undergoing major changes [1,2]. Microbial colonization of the gut starts at birth with the transfer of maternal vaginal, gut, and skin microbiota to the piglet [3]. Moreover, diet and environmental exposures (e.g., housing conditions) are major contributors to subsequent microbial colonization of the gut and development of the immune system [4,5]. It is now recognized that pathogen and environmental exposures impact on the microbial community later in life and that early-life colonization of the mammalian host’s mucosal surfaces plays an important role in the maturation of the host’s immune system [6,7]. For example, many studies in different countries have confirmed that growing up on a farm has a protective effect on asthma and allergies in humans [8]. Another study in mice has shown that low dose treatment with antibiotics can induce obesity later in life. Interestingly, the induction of obesity was not correlated with the antibiotic treatment itself, but with the changes in microbiota [9]. Similar results have been observed for early-life antibiotic treatment in pigs [10]. This implies that changes and interventions in early life, affecting either the microbiota or the immune system, may have long-lasting effects on health, welfare and performance. 

Diet constitutes a pivotal determinant of gut microbial colonization and development, which makes it a compelling target for modulation of the gut microbiota and immune function [11]. Thus, dietary interventions (e.g., prebiotics and probiotics) attracted much interest after the EU wide ban on antimicrobial growth promotors (AMGPs) for production animals in 2006. The notion was that dietary interventions in early life could improve gut health and immune competence, thereby supporting health throughout the vulnerable period post-weaning. 

A relatively well studied dietary intervention in both animals and humans are β-glucans. β-glucans are a group of glucose polymers and are structural components of fungi, algae, bacteria, and plants. Being the subject of numerous studies on their application in animal feed formulations, a lot is known about the molecular mechanisms of β-glucans and their binding to specific cell receptors (e.g., Dectin-1) along the gut lining and specific immune cells, such as antigen-presenting cells (APCs) [12,13]. As reviewed by Thompson et al., several in vitro and in vivo studies showed that β-glucans possess immunomodulatory properties, dependent on their structure, solubility and molecular weight [14]. Especially particulate β-glucans derived from fungi (including yeast), which consist of a (1,3)-β-linked backbone with small numbers of (1,6)-β-linked side chains, are known for their immune-stimulating effects [15,16]. 

Besides the observed immunological effects, there is increasing evidence that β-glucans may have prebiotic traits, and as such might modulate the microbiota [17]. As recently reviewed by Atanasov et al., several fungal polysaccharides, including yeast-derived β-glucans, are already being sold as commercial products due to their high bio-activity [18]. In the last few decades, a small number of studies investigated the effects of yeast-derived products on the intestinal health of pigs [19]. In a recent study that used an autolyzed dried yeast cell product, no effects were shown on caecal microbiota composition [20]. However, it should be noted that that study only investigated the effects of a yeast-derived product in the post-weaning phase.

As outlined above, several studies have investigated the development of the gut microbiota, and the effects of dietary fibres on porcine gut health and the immune function in the post-weaning phase. However, only a few studies have focused their research on the immediate and early period [21]. To the best of our knowledge, no in vivo studies have investigated the effects of yeast-derived β(1,3/1,6)-glucans (MacroGard^®^) on both the porcine gut microbiota and immune system in early life, despite the promising results from several in vitro studies [22,23]. In this study, we investigated whether early-life supplementation of yeast-derived β-glucans affects the gut microbiota as well as the immune system. In addition, we examined the temporal dynamics of these complex systems. We hypothesized that early life consumption of yeast-derived β-glucans alters gut microbiota development (composition) and the immune system. Further developed and mature gut microbiota and immune system in early life could make the animals more resilient against invasive pathogens during the weaning phase and later in life. 

## 2. Materials and Methods 

### 2.1. Ethical Statement

The animal experiment was conducted in accordance with the Dutch law on animal experimentation and ethical requirements, which complies with the European directive 2010/63/EU on the protection of animals used for scientific purposes. The project license application was approved by the Dutch Central Authority for Scientific Procedures on Animals (CCD) (Permit Number: AVD1040020173948) and the animal experiment and associated protocols were approved by the Animal Care and Use Committee of Wageningen University and Research (Wageningen, The Netherlands). Pigs were euthanized by intravenous injection of 20% sodium pentobarbital (Euthasol^®^) followed by immediate exsanguination according to Good Veterinary Practice (GVP). All efforts were made to minimize suffering.

### 2.2. Study Design

To determine the effect of β-glucans on gastro-intestinal development and the immune function in early life, an in vivo study was performed at the Swine Research Centre (Trouw Nutrition, Sint Anthonis, The Netherlands). The experiment was conducted with 33 sows (Hypor Libra, Boxmeer, The Netherlands) and their litters (Maxter × Hypor Libra sow). Sperm from a single boar was used to inseminate all sows to reduce genetic variation in the litters. Shortly after parturition, the piglets received an ear tag, an intramuscular iron injection, their tails were docked, and their birth weight and sex were determined. Calcium carbonate-based powder (Power-Cal^®^, Power-Cal, Sint-Oedenrode, The Netherlands) was added to all pen floors to reduce the infection pressure in the stable. Approximately 24 h after parturition, 96 female piglets were selected and cross-fostered to minimize possible confounding effects such as genetic background, sow parity, litter size, birth weight and time of birth. During cross-fostering, all piglets were randomly allocated to either the control- or treatment (β-glucan) group, while at the same time making efforts to have groups as equal as possible in terms of body weight, sow parity, and time of birth. This resulted in six animals per pen (experimental unit) with 16 pens in total (eight per group) which were divided over four farrowing rooms (balanced for treatment). The complete study timeline and a schematic representation of the experimental design are presented in Figure 1. Non-experimental (male) piglets were equally divided over the pens, with on average six non-experimental animals per pen. All suckling piglets were kept together with their fostering mother until weaning (approximately day 28), each sow being housed in farrowing pens in a room with a computer-controlled climate system. Weaner diet (Appendix A) was administered three days prior to weaning (day 25–27) to allow piglets to become familiar with the consumption of solid feed prior to weaning. At weaning, three pigs per pen (48 pigs in total) were randomly selected and reallocated to a clean nursery facility. During the entire post-weaning period all pens were provided with solid feed and water *ad libitum*. From day 28 until day 44, all pigs received a weaner diet in combination with the feed intervention, and from day 45 until the end of the study the piglets solely consumed a nursery diet (Appendix A). Pigs with a history of medication, leg/claw problems, or growth retardation were excluded from the selection. At day 27, 44, and 70 (dissection), sixteen piglets were sacrificed (*n* = 8 per treatment group, at each time point). 

### 2.3. Experimental Procedures

#### 2.3.1. Dietary Intervention and Oral Vaccination

From day 2 until day 44 of the study, piglets received either β-glucans (MacroGard^®^) or a control treatment (tap water) every other day. Piglets received the dietary interventions orally using disposable syringes (Discardit II, BD). MacroGard^®^ (Orffa Additives B.V., Werkendam, The Netherlands), which consisted of 100% yeast (*Saccharomyces cerevisiae*) cell walls, contains a minimum of 60% particulate β(1,3/1,6)-glucans with an approximate molecular weight of 200 kDa [24]. These β-glucans were not present as soluble molecules but were instead in particulate form as part of yeast cell wall fragments. Especially these (yeast-derived) particulate β-glucans were known for their immune-stimulating effects [15,16]. Other components of MacroGard^®^ were lipids (max. 18%), proteins (max. 8%) and raw ash (max 10%). In addition, the batch did not contain lipopolysaccharides as was assessed in a previous study [22]. The dosage of β-glucans was gradually increased every week with 50 mg, starting from 50 mg per administration in week one to 300 mg per administration in week six. These dosages of β-glucans were chosen by taking previous studies into consideration [25]. To investigate the effect of the yeast-derived β-glucans on the immune function, an oral live attenuated vaccine (Salmoporc^®^ STM (lot number 0270617), IDT Biologika GmbH, Dessau-Rosslau, Germany) was administered on day 21 (primary vaccination) and day 45 (booster vaccination). This vaccine was registered for oral application in pigs to diminish bacterial colonization, excretion, and clinical symptoms of infection with *Salmonella* Typhimurium. The vaccine suspension, containing 5 × 10^8^–5 × 10^9^ CFU/mL of the live attenuated *Salmonella* enterica serovar Typhimurium, was freshly prepared according to manufacturer’s instructions prior to oral administration. 

#### 2.3.2. Blood and Faecal Sampling 

Blood and faecal samples were collected at different time points during the study for the evaluation of immune function and microbiota composition, respectively. Faecal samples were collected on days 4 (*n* = 24 per treatment group), 8 (*n* = 24), 14 (*n* = 24), 26 (*n* = 24), 35 (*n* = 16), 43 (*n* = 16), 59 (*n* = 8), and 69 (*n* = 8). Rectal stimulation was performed by inserting the tip of a wetted (with sterilized H_2_O) cotton swab (PurFlock Ultra, Puritan) into the rectum, by making small, gentle movements (circular and back- and forward). These fresh faecal samples were collected in cryotubes, immediately placed on dry ice and stored at −80 °C until further processing. DNA isolation from faeces was done on all faecal samples from animals that were dissected. Blood was collected on days 14, 26, 43, and 69 from the jugular vein of the pig using Natrium Heparin tubes (S-monovette^®^, Sarstedt, Germany). Blood samples were either kept at room temperature (RT) until further processing for cell analysis or centrifuged at 2000× *g* for 10 min to collect serum. Serum was stored at −20 °C until further use. Piglets were sampled in a random order and weighed directly before sample collection. 

#### 2.3.3. Dissection 

After euthanasia and exsanguination, the ileocecal mesenteric lymph node (MLN) was removed and stored on ice-cold (4 °C) RPMI 1640 Medium (with GlutaMAX™ supplement, Gibco^®^), containing 10% fetal calf serum (FCS, Gibco^®^) and 1% L-Glutamine (Gibco^®^). Subsequently, the GI tract was removed from the abdominal cavity of the piglet, and the jejunum, ileum, and caecum were identified and segmented accordingly. Digesta samples from these GI segments were taken on day 27 (pre-weaning), and day 44 and 70 (post-weaning) by gently squeezing the segment content into a plastic container. Next, the collected digesta was completely homogenized using a clean spatula. Approximately 1 g of homogenized digesta was stored in a sterile cryogenic vial, snap-frozen on dry ice and stored at −80 °C until further processing. The remainder of the digesta from each GI segment was mixed with H_2_O for pH measurement (Appendix A) using a pH meter (ProLine B210).

### 2.4. Measurements

#### 2.4.1. Microbiota Analysis

DNA was extracted from faecal and digesta samples using a customized Maxwell 16 Total RNA protocol (Promega Corp., Madison, WI, USA) with Stool and Transport and Recovery Buffer (STAR; Roche Diagnostics Corp., Rotkreuz, Switzerland). Briefly, 50 mg of previously frozen (−80 °C) faeces or 100 mg of digesta was homogenized with 0.25 g of sterilized 0.1 mm zirconia beads and three 2.5 mm glass beads in 300 μL STAR buffer for 3 × 1 min at 5.5 m s^−1^ using a bead beater (Precellys 24, Bertin Technologies, Montigny-le-Bretonneux, France), with a waiting step of 15 s in between. Samples were incubated with shaking at 300 rpm for 15 min at 95 °C and centrifuged for 5 min at 4 °C and 16,100× *g*. The supernatant was removed and the pellets were processed again using 200 μL fresh STAR buffer. Samples were incubated at 95 °C and centrifuged as before. The supernatant was removed, pooled with the first supernatant and 250 μL was used for purification with Maxwell 16 Tissue LEV Total RNA Purification Kit, catalogue no. AS1220 (Promega Corp.) customized for DNA extraction in combination with the STAR buffer. DNA was eluted with 50 μL of DNAse- and RNAse-free water (Qiagen). DNA concentrations were measured with a NanoDrop ND-1000 spectrophotometer (NanoDrop Technologies Inc., Wilmington, DE, USA) and adjusted to 20 ng μL^−1^ with DNAse- and RNAse-free water. PCR amplification was carried out with barcoded primers directed to the V4 region of the bacterial and archaeal 16 S rRNA gene, namely EMP_515 F (5′-GTGYCAGCMGCCGCGGTAA), with linker ‘GT’, and EMP_806 R (5′-GGACTACNVGGGTWTCTAAT), with linker ‘CC’. PCR reactions were done in duplicate, each in a total volume of 50 μL and containing 20 ng of template DNA. In order to distinguish samples between sequencing reads, each sample was amplified with a unique barcoded primer pair (200 nM each per reaction), 1 × HF buffer (Thermo Fisher Scientific, Waltham, MA, USA), 1 μL dNTP Mix (10 mM each; Roche Diagnostics GmbH, Baden-Württemberg, Germany), 1 U Phusion Hot Start II High Fidelity DNA Polymerase (Thermo Fisher Scientific), and 36.5 μL of DNAse- and RNAse-free water. The amplification program included 30 s initial denaturation at 98 °C, followed by 25 cycles (with the exception of jejunum digesta samples which were processed with 30 cycles to yield sufficient amplicon fragments) of denaturation at 98 °C for 10 s, annealing at 56 °C for 10 s, elongation at 72 °C for 10 s, and a final extension at 72 °C for 7 min. The PCR product presence and size (≈290 bp) was confirmed with gel electrophoresis using a 1% agarose gel. In each library, 69 unique barcode tags were used and two artificial (mock) communities were included in addition to a water (blank) control [26]. PCR products were purified using the HighPrep PCR kit (MagBio Genomics Inc., Gaithersburg, MD, USA), and DNA concentrations were measured using the Qubit dsDNA BR Assay Kit (Thermo Fisher Scientific). Of each barcoded sample, 200 ng DNA was added to an amplicon pool that was subsequently concentrated using the HighPrep PCR kit to a volume of 20 μL. The DNA concentration of the amplicon pool was measured using the Qubit dsDNA BR Assay Kit and the libraries were sent for Illumina HiSeq sequencing (Eurofins Genomics, Ebersberg, Germany). Amplicon sequence data were processed and analysed using NG-Tax 2.0 [27] and annotated using the SILVA 132 database [28].

#### 2.4.2. Serology

To analyse the antibody response against *Salmonella* Typhimurium, an in-house ELISA was optimised and performed as described previously [29]. In short, Salmoporc^®^, containing a live attenuated strain of *Salmonella enterica* serovar Typhimurium, was streaked on a MacConkey agar (Sigma Aldrich) plate and grown overnight at 37 °C. Then, a single colony was used to inoculate 2 mL of lysogeny broth (LB) medium, and grown overnight under aeration on a shaker (200 rpm) at 37 °C. The following day, 1 mL of the overnight culture was transferred to a 50 mL tube containing 15 mL of LB medium. The 50 mL tube was placed under aeration on a shaker (200 rpm) at 37 °C until reaching the exponential growth phase (OD_600 nm_ = 0.6–0.8). Thereafter, the bacteria were pelleted by centrifugation at 10,000 rpm for 5 min and washed two times with cold PBS. Medium-binding 96 well plates (clear, flat bottom, Greiner Bio-One, Vilvoorde, Belgium) were coated with 100 µL of the bacterial suspension (2 × 10^8^ bacteria/mL) and incubated overnight at 4 °C. The following day, non-attached bacteria were removed and attached bacteria were fixed with 4% paraformaldehyde for 2 h at RT. Plates were blocked overnight at RT with a blocking solution consisting of 5% milk powder (ELK, FrieslandCampina, Amersfoort, The Netherlands) in demi water, and stored at 4 °C until usage. After blocking, plates were washed with PBS/Tween20 (0.05%) and 100μL of diluted serum samples were added. Serum samples were diluted 250 × (IgG) and 50 × (IgA, IgM) in blocking solution, and incubated for 1 h at RT. Plates were washed two times, and 100 μL of 50,000 times diluted (in blocking solution) horseradish peroxidase (HRP) conjugated goat anti-Porcine IgG, IgM or IgA (Novus Biologicals, Centennial, CO, USA) was added. After 30 min, plates were washed five times and incubated with 100 µL of 3,3′,5,5′-tetramethylbenzidine (TMB) substrate solution (Enhanced K-Blue^®^, Neogen, Lansing, MI, USA). After 15 min the reaction was stopped with 100 µL of a 2% HCl stop solution, and the optical density (OD) of the plates was measured at a wavelength of 450 nm (Multi-Mode Microplate Reader FilterMax F5). 

#### 2.4.3. Cell Isolation 

Collected blood was diluted 1:1 with PBS (containing 0.5 mM EDTA) within 4 h of collection and transferred to Leucosep^®^ tubes using a 60% FICOLL-PAQUE™ Plus density-gradient to isolate the peripheral blood mononuclear cells (PBMCs). Remaining red blood cells were lysed with ACK lysis buffer (Gibco^®^). MLN cells were collected by cutting the nodules in smaller pieces, and gently squeezing and flushing the cells through a sterile Falcon**^®^** cell strainer (100 µm, Corning^®^, New York, NY, USA), using a syringe plunger and sterile Mg^2+^ and Ca^2+^ free PBS (Lonza, Basel, Switzerland). Both isolated PBMCs and MLN cells rested overnight at 4 °C until further processing. 

#### 2.4.4. Stimulation Assay

After resting overnight, 1 × 10^6^/well of PBMCs and MLN cells were separately transferred to a 96 well clear round bottom plate (Greiner Bio-One). Plates containing wells with LPS (serotype O55:B5/L2880, Sigma-Aldrich, St. Louis, MO, USA), with Concanavalin A (Con-A, C2010, Sigma-Aldrich), or without a stimulus (cell culture medium only) were prepared the previous day and stored at 4 °C overnight. The next day, the plates were placed at 37 °C for 30 min before addition of the cells. Cells were incubated with 0.1, 1 or 10 µg/mL of LPS, with 5, 2.5, or 1.25 µg/mL or without stimuli for 24 h at 37 °C (5% CO_2_). After 24 h of incubation, the plates were centrifuged at 300× *g* for 3 min to spin the cells down. Subsequently, the cell culture supernatant was collected and immediately stored in 96 well polypropylene plates (Nunc^®^, MicroWell^TM^, Sigma-Aldrich) at −80 °C until further processing.

#### 2.4.5. Flow Cytometry

Isolated PBMCs and MLN cells were stained with a DC or T lymphocyte/NK cell antibody panel (Table 1). To analyse these cells, 5 × 10^6^ (DC panel) or 1 × 10^6^ (T lymphocyte/NK) freshly isolated PBMCs or MLN cells (per animal) were placed in a 96 well polypropylene plate (Nunc^®^, MicroWell^TM^, Sigma-Aldrich). Per well, 200 µl FACS buffer (Mg^2+^ and Ca^2+^ free PBS; Lonza), 2 mM EDTA (Merck, Kenilworth, NJ, USA), 0.5% BSA fraction V (Roche, Basel, Switzerland) was added and the plates were centrifuged at 400× *g* for 3 min at 4 °C to wash the cells. After washing, extracellular cell surface markers (Table 1) were stained with antibodies for 30 min on ice (in the dark), followed by three washing steps with cold PBS. Subsequently, cells were stained with (in PBS diluted) Fixable viability dye eFluor^TM^ 506 (eBioscience^TM^) and Streptavidin-BV421 (DC panel) for 30 min, followed by a washing step with FACS buffer. Then, 100 μL Fix/Perm buffer (eBioscience^TM^) was added to each well and incubated for 45 min at RT, to allow for intracellular staining. After incubation, cells were washed three times in Perm buffer (eBioscience^TM^), followed by the incubation with the intracellular antibody mix (Table 1) in 35 μL Perm buffer for 30 min at 4 °C. After two other washing steps with Perm buffer, cells were resuspended in 200 μL FACS buffer and measured for 150 s (NK cells/T cells) or 300 s (DCs) on the FACS CANTO II at a medium flow rate. Beads were used for single stained controls using compensation beads (UltraComp eBeads^TM^, ThermoFisher), and cells were used for Fluorescence Minus One (FMO) controls to control for spectral overlap. Flow cytometry data analysis was performed by using FlowJo^TM^ software (Version 10). Gating of DC subsets, NK cells and T cell subsets was performed in line with previous studies [30,31,32]. Examples of the gating strategies can be found in Appendix A (DCs) and Appendix A (T lymphocytes and NK cells).

### 2.5. Statistical Analysis

All R-scripts, data files and pdf files with extensive information on the performed analyses can be accessed through the following DOI: 10.4121/12999620. Alternatively, these files can be found under the following Github page: https://github.com/mibwurrepo/de-Vries-et-al-2020-porcine-study-beta-glucans.

#### 2.5.1. Microbiota Analysis

In order to estimate the impact of yeast-derived β-glucans on alpha diversity of the microbiota in piglets’ faeces and digesta, Shannon and InvSimpson diversities were calculated for each sample. Shannon and InvSimpson diversities were both chosen as Shannon’s diversity index gives more weight to rare species and Simpson’s index gives more weight to common species. It should be noted that really rare sequences are missing from the dataset as NG-Tax 2.0 uses an abundance threshold of 0.1% for amplicon sequence variants (ASVs) in a given sample. Shannon and InvSimpson diversities were used in a Linear Mixed-Effects Model (i.e., Shannon~Day_of_study * Treatment) to test for significant differences between time points and treatment groups (nlme package v3.1-140) [33]. To estimate the impact of β-glucans on overall microbiota composition, PERMANOVA tests were performed using Bray-Curtis dissimilarities at ASV level (9999 permutations). The homogeneity assumption was tested by calculating Bray-Curtis dispersion for each group followed by ANOVA tests. To visualize the beta diversity in faecal samples over time, a Principal Coordinates Analysis (PCoA) plot was generated using weighted Unifrac dissimilarities. In order to estimate the impact of β-glucans as a dietary intervention on the relative abundance of specific genera in piglet faeces, reads of the ASVs were first aggregated to genus level using the tax_glom function of the phyloseq R package [34], after which read frequencies were transformed to compositional data using the transform function of the microbiome R package [35]. Genera were filtered to exclusively include genera that had a relative abundance of 0.1% in at least 50% of the samples. Resulting genera were tested for differential relative abundance using the Generalized Additive Models for Location, Scale and Shape (GAMLSS) with a zero-inflated beta family (function taxa.compare) with GAMLSS-BEZI as a statistical method in the metamicrobiomeR package [36]. Dietary intervention was used as the main variable for comparison and day of study and department (farrowing rooms) were used as adjusting variables. Ear tag was indicated as the identifier to enable a longitudinal approach and “fdr” (False Discovery Rate) was chosen as the method for multiple testing adjustment (*p*.adjust < 0.05 as the threshold). This analysis was performed on three different datasets: a dataset containing only pre-weaning faecal samples, one with only post-weaning faecal samples and one dataset containing faecal samples from all time points (both pre- and post-weaning). To get a better overview of the genera that mostly contributed to the differences in microbiota composition between the treatment groups over time, weighted Unifrac distance-based Principal Response Curve (dbPRC using package Vegan [37,38]) analysis was performed. For these dbPRC figures, pre- and post-weaning datasets were used to investigate these distinct periods in the piglet’s microbiota development. 

#### 2.5.2. Immunological Analysis

To assess if yeast-derived β-glucans alters the immune system, a Student’s t-test (unpaired) was performed using R statistical software (version 3.6.2). Additionally, a Two-way ANOVA followed by a Tukey post-hoc test was performed to assess the effects over time and the interaction between time and treatment. Normality of data (Shapiro-Wilk test) and homogeneity of variances (Levene’s test) were checked prior to statistical testing. Skewness values between −2 to +2 were considered acceptable [39]. Extreme outliers (indicated by R) were removed from the analysis. When the statistical assumptions were not met, data were log transformed. Data are presented as untransformed means ± SEM. Results with an adjusted *p*-value below 0.05 (*p* < 0.05) were considered statistically significant and results between 0.05 and 0.1 (0.05 < *p* < 0.10) were considered a trend. Significant results were tested for confounding factors, including body weight at birth, using an ANCOVA. 

#### 2.5.3. Correlation Analysis

To assess possible correlations between microbial composition, study factor- and immunological data, first microbiota ASVs were aggregated at the genus level and transformed to compositional abundances. Then, a prevalence filter was applied with a 0.1% abundance threshold in at least 50% of the samples. Resulting genera were combined with study factor data and immunological data and a test for association was performed for each parameter pair, using a Pearson moment correlation (95% CI). *p*-values were corrected for multiple testing using the ‘Benjamini-Hochberg’ method. 

## 3. Results

### 3.1. Microbial Colonization

#### 3.1.1. Alpha Diversity 

To assess potential differences in microbiota composition between β-glucan treated animals and control animals, both faecal and luminal samples were collected at multiple time points (Figure 1). No significant differences for alpha diversity were observed between the treatment groups when including all faecal time points (Figure 2; *p* = 0.11 for Shannon, *p* = 0.60 for InvSimpson). However, when including exclusively pre-weaning faecal time points, a significant difference in alpha diversity between the treatment groups was observed for the Shannon diversity (*p* < 0.04 for Shannon, *p* < 0.10 for InvSimpson). The significant difference for Shannon’s diversity indicates that low abundance ASVs are mostly contributing to the differences in alpha diversity. As expected, piglet faecal and luminal microbiota composition showed different temporal trends in alpha diversity, with alpha diversity in faeces increasing up to day 35 before reaching a plateau that lasted until the end of the study. Furthermore, digesta from different gut segments all showed a similar pattern with a higher α-diversity on day 27 in comparison to day 44 (*p* < 0.001).

#### 3.1.2. Microbiota Composition over Time

The dynamic shifts in the microbiota composition in the faeces and digesta over time were examined at the phylum and family classification level. At the phylum level (Figure 3A), the microbiota composition in faeces transitioned from a *Firmicutes*, *Bacteroidetes*, *Proteobacteria,* and *Fusobacteria* predominated community on day 4 to a *Firmicutes* and *Bacteroidetes* predominated community at later time points. Both the jejunum and ileum digesta were characterized by a *Firmicutes* predominated community at all time points (day 27, 44, and 70), whereas the caecal microbiota was found to be predominated by *Firmicutes* and *Bacteroidetes* at all time points. Furthermore, the average relative abundance of the *Euryarchaeota* in faeces (Appendix A) increased over time during the pre-weaning period from below 0.03% on day 4 to reach an average relative abundance of over 6% on day 26. Post-weaning faecal samples show a more stable pattern of the *Euryarchaeota* with a relative abundance of 1–2% on each time point until day 69. Interestingly the *Euryarchaeota* were present in caecal digesta with a similar abundance (1.5%) on day 27, while at later time points the mean relative abundance in caecal digesta was drastically lower (<0.1% at both day 44 and day 70). In jejunum and ileum digesta, the relative abundance of the *Euryarchaeota* was very low at all time points, except for jejunum digesta on day 70 (Appendix A). At the family level (Figure 3B), a dramatic shift of taxa could be observed in faeces over time, with large shifts occurring from day 4 to day 8 and during the weaning transition, from day 26 to day 35. Jejunum digesta was predominated by *Lactobacillaceae* at all time points, while ileum digesta was predominated by *Lactobacillaceae* at day 27 and day 44 followed by *Clostridiaceae*_1 on day 70. Caecum digesta was predominated by *Lactobacillaceae, Ruminococcaceae,* and *Prevotellaceae* on day 27, by *Lactobacillaceae* and *Prevotellaceae* on day 44 and by *Prevotellaceae* on day 70. For a more detailed overview of the relative abundance of several families in faecal samples over time, see Appendix A.

#### 3.1.3. Beta Diversity

Beta diversity was calculated at the ASV level using Bray-Curtis dissimilarities and a PERMANOVA test was performed on all faecal samples with treatment and time point as model parameters. The effect of treatment was not significant (*p* > 0.14), but a strong time effect was observed (*p* < 0.001). When including only pre-weaning faecal samples, however, the effect of treatment was significant (*p* < 0.042), but the contribution of treatment to the variance was low (R^2^ < 0.0031). This was not observed when only post-weaning samples were included. Bray-Curtis dispersions of pre- and post-weaning faecal samples were compared and showed to be significantly different (ANOVA, *p* < 0.001), with dispersion being lower in post-weaning faecal samples. Principal Coordinates Analysis (PCoA) based on weighted Unifrac distance of the piglet faecal microbiota revealed significant temporal shifts and a distinct separation between pre- and post-weaning faecal samples (Figure 4). The wide distribution of the samples collected on days 4, 8, and 14 reinforced the notion that the overall microbiota composition of piglet faeces changes dramatically in the first two weeks after birth. 

#### 3.1.4. Differentially Abundant Genera between Treatment Groups

When comparing the relative abundances of individual taxa in pre-weaning faecal samples at the genus level, three out of a total of 31 genera that passed the prevalence filter were found to be differentially abundant between the β-glucan and control group (Figure 5). Especially on days 8 and 14, the genus *Methanobrevibacter* was less abundant in animals that received β-glucans, while the genera *Fusobacterium* and *Ruminococcaceae*_UCG-002 were more abundant at a number of time points in animals that received β-glucans. No differentially abundant genera were found in faecal samples of the post-weaning period, nor were differentially abundant genera found when all faecal samples were used in the GAMLSS model. The dbPRC analysis (Appendix A) revealed several additional genera that contributed to the changes between treatment groups over time.

### 3.2. Immunological Analysis

#### 3.2.1. Immunological Response to Oral Vaccination 

To analyse if supplementation of β-glucans in early life induces changes to the host’s innate and adaptive immune system, different types of analysis were performed to assess this on a local and systemic level. In this study, an oral vaccination against *Salmonella* Typhimurium was administered to measure whether yeast-derived β-glucans have an effect on the responsiveness of the immune system. As expected, a clear and significant increase of *Salmonella*-specific IgM and IgA antibodies could be observed after both the primary and the secondary (booster) vaccination (Figure 6A,B). An increase in antigen-specific IgG antibodies was only observed after the second vaccination (Figure 6C). Despite the clear increase of antibody titres, no altered antibody response to the *Salmonella* vaccine strain was observed between the β-glucan and control group during the pre- and post-weaning phase of the study (Figure 6A–D).

#### 3.2.2. Ex Vivo Stimulation of MLN Cells and PBMCs 

Figure 7 shows the production of the anti-inflammatory cytokine IL-10 by MLN cells and the production of the pro-inflammatory cytokine TNFα by PBMCs, upon stimulation with LPS or Con-A. The cytokines were measured at three different time points (one pre- and two post-weaning) during the study. Significant differences were only observed pre-weaning. On day 27, MLN cells from β-glucan treated animals produced higher levels of IL-10 upon stimulation with 10 µg/mL LPS (*p* = 0.008; T-test) in comparison to LPS stimulated MLN cells from non-treated animals (Figure 7A). Using a lower concentration of LPS (1 µg/mL) showed the same pattern but resulted in a lower level of IL-10 production (Appendix A). In accordance with these results, Con-A stimulated MLN cells from β-glucan treated animals showed an increased IL-10 production on day 27 (Figure 7C), albeit not significant (*p* = 0.09; T-test). No differences in IL-10 production were observed for stimulated PBMCs (Appendix A). Interestingly, MLN cells from β-glucan treated animals also produced more IL-10 when left unstimulated (medium only) compared to unstimulated MLN cells from the control animals (*p* = 0.057; T-test). No such differences were observed for the post-weaning time points (day 44 and 70, respectively) and the variation between individual animals increased in the post-weaning phase. No differences in TNFα production by MLN cells were observed between the treatment groups (Appendix A), but stimulation of PBMCs obtained at day 27 from β-glucan treated animals showed a decreased TNFα production upon stimulation with Con-A (*p* = 0.024; T-test) and LPS (not significant) (Figure 7B,D). Furthermore, IL-10 production changed over time as observed for Con-A stimulated MLN cells (Figure 7C), independent of the different treatment groups. As for LPS stimulated PBMCs (Figure 7C), a significant increase of TNFα was only observed for β-glucan treated animals (time-treatment interaction). 

#### 3.2.3. Analysis of Immune Cells in Blood and the MLN

Cell analyses of PBMCs and MLN cells were used to investigate the potential effects of β-glucans on the abundance (percentage of cells) or function (maturation or proliferation) of different immune cell populations, including DCs, NK cells and T cells (Table 2: PBMC, Table 3: MLN). No significant differences were observed for the DC subsets in PBMCs at any of the time points. However, a decreased percentage of proliferating NK cells (*p* = 0.009; T-test, *p* = 0.010; Two-way ANOVA) and γδ T cells (*p* = 0.049; T-test, *p* = 0.034; Two-way ANOVA) was observed for β-glucan treated animals on day 26 (Figure 8A,D). As for MLN cells, the same immune cell populations were analysed, with the exception of NK cells. On day 44, a higher percentage of cDC1 (*p* = 0.011; T-test) and cytotoxic (CTL) cells (*p* = 0.035; T-test) was observed in β-glucan treated animals. In contrast, a lower expression of the cell maturation marker CD80/86 was observed for pDCs on day 27 (*p* = 0.024; T-test) and 70 (*p* = 0.034; T-test). Interestingly, correlation analysis showed that the *Romboutsia* genus positively correlates with the expression of the cell maturation marker CD80/86 on cDC1 cells (Appendix A). Furthermore, a significant time effect was observed for almost all analysed cell populations (Table 2 and Table 3), generally with clear differences between day 14 and day 26 (pre-weaning) and between the pre- and post-weaning time points. 

## 4. Discussion

The microbiota plays an important role in in the development of the innate and adaptive immune system [40]. However, the interplay between diet, the gut microbiota, and the immune system is still poorly understood, especially in early life. In this study, we investigated the impact of yeast-derived β-glucans on the porcine gut microbiota and the immune system from birth to six weeks post-weaning. We hypothesized that early-life supplementation of yeast-derived β-glucans (MacroGard^®^) alters gut microbial development and potentiates immunological responses in pigs. Overall, supplementation of yeast-derived β-glucans had no profound effect on the development of the gut microbiota and the immune system, and any observed differences were modest. However, clear time effects were observed in the gut microbiota and the immune system between the pre- and post-weaning time points, which gives valuable information regarding the development of these complex systems. Importantly, no adverse effects of orally administered β-glucans were observed, and there were no indications that β-glucans had a negative effect on physiological parameters (e.g., body weight and pH), gut microbiota, or the immune system in early life.

Yeast-derived β-glucans are being used as functional ingredients in pig nutrition to boost gut health. Although β-glucans have been shown to impact the immune system by binding to specific host receptors, β-glucans can also be considered as dietary fibres. Dietary fibres can potentially be used to impact gut microbiota composition by stimulating the growth of saccharolytic or fibrolytic microorganisms [41]. Similarly, β-glucans may, depending on solubility, escape digestibility in the upper GI tract and may arrive in the lower GI tract where they are broken down by saccharolytic bacteria into short-chain fatty acids (SCFAs) [42]. A meta-analysis on the impact of the inclusion of cereal-derived β-glucans in feed has indeed shown a positive correlation with the presence of the SCFA butyrate in the colon [43]. In the present study, three genera were found to be differentially abundant over time in faeces during the pre-weaning phase when given yeast-derived β-glucans as dietary intervention, including *Methanobrevibacter*, *Fusobacterium,* and a genus within the family of *Ruminococcaceae.* Interestingly, in a porcine study in which dietary fibres were administered during the pre-weaning period (day 14 until weaning), effects were seen on the relative abundance of several genera in the mid-colon, amongst which two genera in the *Ruminococcaceae* family [44], highlighting the potential of microbiota modulation by dietary fibres. In a study where the effects of three purified β-glucans were tested on members of the GI tract, all three were shown to reduce the *Enterobacteriaceae* population [45]. In the current study, this shift in *Enterobacteriaceae* was, however, not observed. 

Besides the differentially abundant genera, minor but significant differences in alpha and beta diversity were found in faeces between treatment groups in the pre-weaning period. These differences were not found in digesta from jejunum, ileum or caecum, which might be explained by the smaller sample size. As expected, a clear temporal pattern was observed for both treatment groups. This includes a clear separation between pre- and post-weaning time points as demonstrated before in similar studies [46,47,48]. The alpha diversity increased in the early pre-weaning phase (day 4) up to the first post-weaning time point (day 35), after which a plateau was reached. This is in line with previous studies [46,47]. In another study, alpha diversity reached a plateau from day 24 onwards [21]. This outcome likely has to do with the earlier time of weaning (day 21), demonstrating the dominant role that the weaning transition has on gut microbiota maturation. The GI tract’s microbiota composition found in the present study is comparable to that observed in other studies in terms of main phyla in faeces and caecum, both pre- and post-weaning [49,50,51]. In addition, main phyla found in the jejunum and ileum post-weaning were comparable to the results from another study [52]. In addition, a study by Kim et al. found that the *Prevotellaceae* family is predominant at ten weeks of age [50], which was also observed in our study. Moreover, the temporal pattern of the composition and relative abundance of archaeal taxa is comparable with data from multiple studies [49,53].

Both the composition and development of the microbiota as well as the innate and adaptive immune maturation are influenced by exposure to commensal microorganisms and dietary antigens [4]. Dietary β-glucans are recognized by pathogen recognition receptors, the most important one being Dectin-1 [54]. Sonck et al. demonstrated that Dectin-1 is expressed in the spleen, MLN, lungs, and along the digestive tract of pigs [55]. Immune cells, including DCs, are also known to express Dectin-1 in human and mice, but this has not yet been shown for porcine immune cells [56,57]. Recognition of β-glucans by immune cells may induce altered immune responses upon a secondary stimulation, thereby influencing the immune competence of the pig. To test this, we assessed if pigs treated with β-glucan showed an enhanced immune response to a live attenuated *Salmonella* vaccine (Salmoporc^®^). A clear vaccine-induced increase in *Salmonella* specific antibodies was observed, but no differences were found between the control and β-glucan group. This could imply that the vaccination induced an immunological response that was too strong to determine the subtle effects of this dietary intervention on the systemic immune competence, or that yeast-derived β-glucans do not enhance the systemic immune system. In line with these results, we did not see major differences in the percentage and proliferation of T-cell subsets in both blood and the MLN. These results are in accordance with a previous study that investigated the effect of (purified) yeast-derived (1,3/1,6)-β-d-Glucan (Wellmune WGP^®^) on the vaccination response and T-cell subsets in neonatal piglets. The authors did not find any effect of this dietary supplement on intestinal or immune development and it did not improve the antibody response to vaccination in neonatal piglets [58]. 

Interestingly, when we stimulated PBMCs from β-glucan treated animals ex vivo with LPS or Con-A, we observed a reduction in the production of the pro-inflammatory cytokine TNFα on day 26. In line with these findings, ex vivo stimulated MLN cells from β-glucan treated animals produced higher levels of the anti-inflammatory cytokine IL-10 on day 27. These results suggest that yeast-derived β-glucans induce a more anti-inflammatory or tolerant state of the systemic immune system. This is also suggested by the reduced percentages of CTLs and proliferating NK cells and γδ T cells in PBMCs from β-glucan treated animals on day 26. The observed reduction of TNFα in β-glucan treated animals has been observed in other porcine in vivo studies. Vetvicka et al. showed a significant reduction of TNFα in blood serum after an endotoxin challenge with LPS [59]. In addition, Li et al. observed a decrease of TNFα production by blood lymphocytes from β-glucan treated pigs which were stimulated ex vivo with LPS [60]. Taken together, these findings suggest that β-glucans may alter immune cell responses upon a secondary stimulus as was observed in a study by dos Santos et al. [61], in which they showed a significant increase in IL-10 production after 2 and 4 h of infection with *L. braziliensis* in β-glucan trained macrophages. The modest effects of yeast-derived β-glucans on the immune system were primarily observed in the pre-weaning phase, which is in line with the findings from the microbiota analysis (Figure 2 and Figure 3). A more complex diet after weaning may impair the ability to identify any subtle changes induced by yeast-derived β-glucans in the post-weaning period.

Most studies suggest that yeast-derived β-glucans improve the overall health of pigs. However, strong and consistent evidence is lacking, and no studies have investigated its impact on both the gut microbiota and the immune system. The present study shows that yeast-derived β-glucans primarily induce effects in the gut microbiota and immune system early in life. During the pre-weaning period, the diversity of the faecal microbiota was lower in animals that received β-glucans. This is unexpected, as fibrous components typically result in increased microbial diversity [62]. As previously discussed, three genera were found to be differentially abundant in the same period, including *Methanobrevibacter*, *Fusobacterium,* and a genus within the *Ruminococcaceae* family. At the genus level, it is hard to state whether these are positive effects, as a species within these genera can either act in a commensal or beneficial fashion, or can have adverse effects for the host. Interestingly, the genus *Romboutsia* showed a positive correlation with the expression of the cell maturation marker CD80/86 on cDC1 cells (Appendix A). This may imply that *Romboutsia* contributes to the maturation of the (innate) immune system. Several *Romboutsia* spp. are commensals of the mammalian GI tract and have been shown to be flexible anaerobes that are adapted to the small intestine [63,64]. As for the immune system, the increased production of the anti-inflammatory cytokine IL-10 by MLN cells, and the decreased production of TNFα by PBMCs, suggest an anti-inflammatory effect. However, it should be noted that only a small number of animals were sampled and that the level of LPS induced IL-10 produced by MLN cells was relatively low (0–61 pg/mL). Nevertheless, these results are in line with other in vivo studies that showed limited or no effects of yeast-derived β-glucans on the (mucosal) immune system or vaccination response [58,65,66,67]. 

Taken together, this study reinforces that the pre-weaning period constitutes a window of opportunity to alter the immune system through diet, which may set the stage for immune homeostasis and subsequent host–microbial interaction. However, the observed changes induced by dietary β-glucans were modest, and any long-lasting health effects of yeast-derived β-glucans remain to be elucidated. 

## Figures and Tables

**Figure 1 microorganisms-08-01573-f001:**
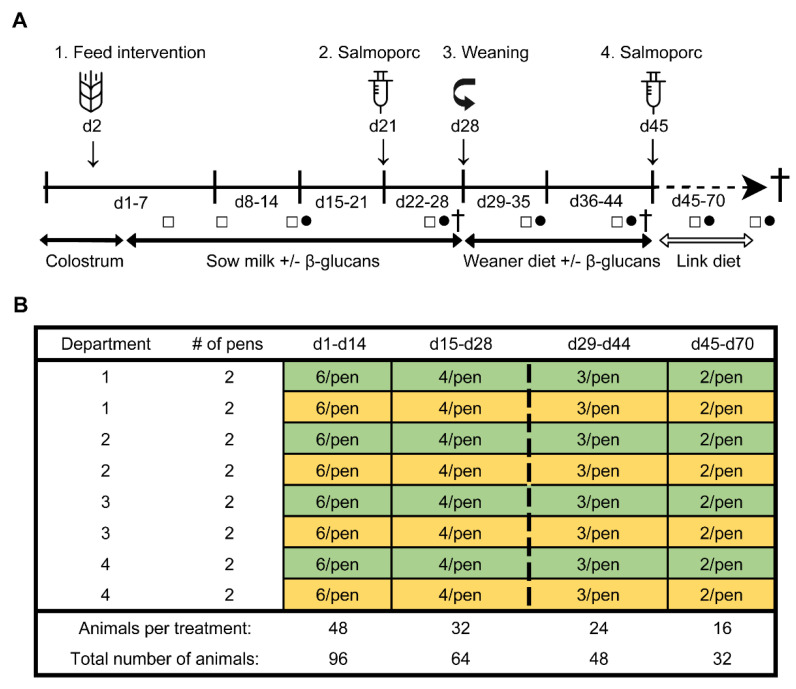
Study timeline (**A**). From day 2 until day 44, all piglets orally received either β-glucans or a control treatment (tap water) every other day (1). A sub-selection of piglets was weaned on day 28 (3). Pigs were vaccinated with Salmoporc^®^ on day 21 (primary vaccination; (2) and on day 45 (booster vaccination; (4). Faecal samples were collected on days 4, 8, 14, 26, 35, 43, 59 and 69 (squares; □). Blood samples were taken on days 14, 26, 35, 43, 59, and 69 (circles; ●). A subset of animals were sacrificed on days 27, 44, and 70 (cross; †). Schematic representation of the experimental design showing the remaining number of animals per treatment over time as determined by deselection or dissection of animals (**B**). The control and the β-glucan groups are presented as yellow and green, respectively. The dashed line separates the pre-weaning and post-weaning time points.

**Figure 2 microorganisms-08-01573-f002:**
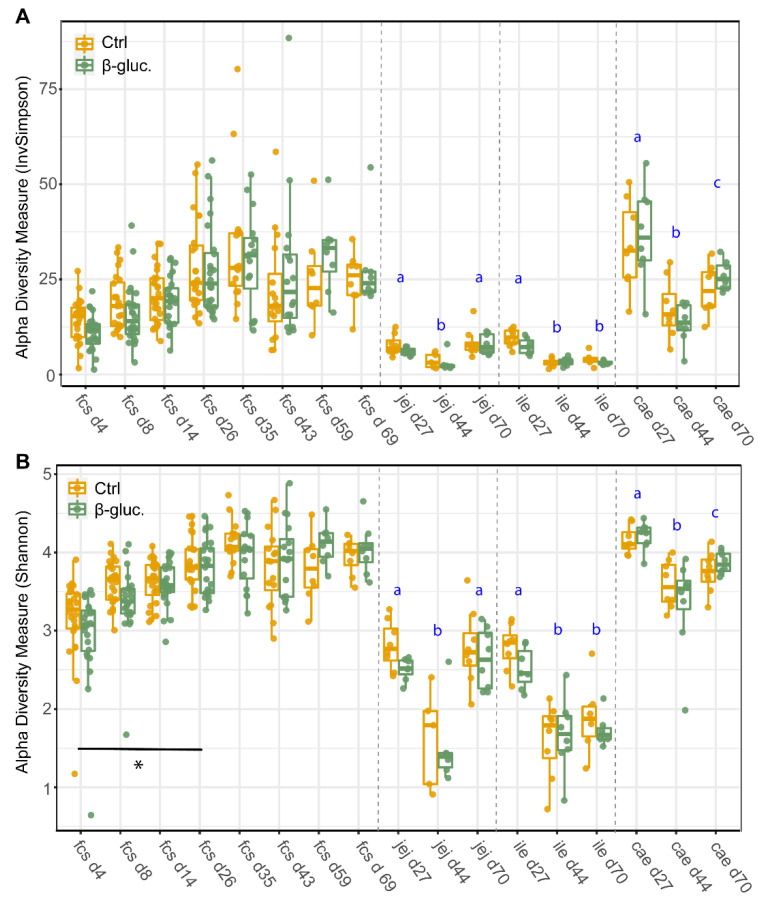
Alpha-diversity of faecal and luminal microbiota between treatments and over time. Every dot represents a single animal with the control animals in red and the β-glucan treated animals in green. InvSimpson diversity values (**A**) and Shannon diversity values (**B**) are given by sampling time point (d4-70) and by faeces (fcs) or gut segment; jejunum (jej), ileum (ile), caecum (cae). Letters (a, b, c) represent significant differences (*p* < 0.05) between time points. The horizontal line with the corresponding asterisk represents the significant difference between treatment groups in the pre-weaning period, calculated using a Linear Mixed-Effects Model (*; *p* < 0.05).

**Figure 3 microorganisms-08-01573-f003:**
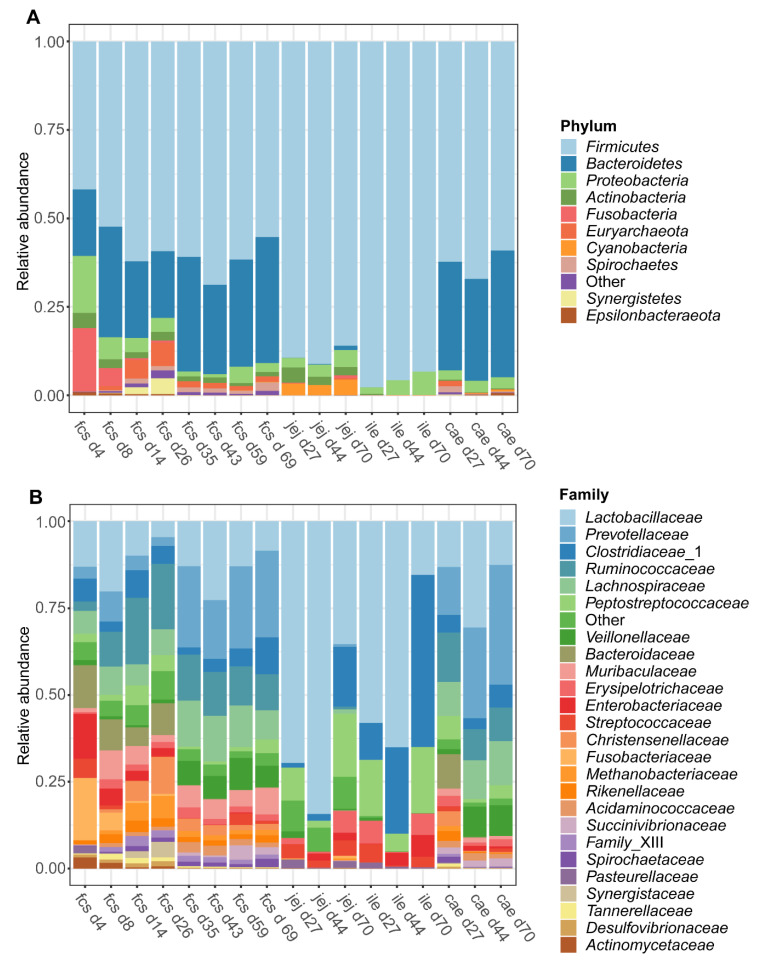
Taxonomic composition of the piglet faeces and digesta over time at phylum (**A**) and family (**B**) level. Data are given as the mean relative abundance by sampling time point (d4-70) and by faeces (fcs) or gut segment; jejunum (jej), ileum (ile), caecum (cae). Data includes samples from both treatment groups and shows the 25 most abundant families and 10 most abundant phyla.

**Figure 4 microorganisms-08-01573-f004:**
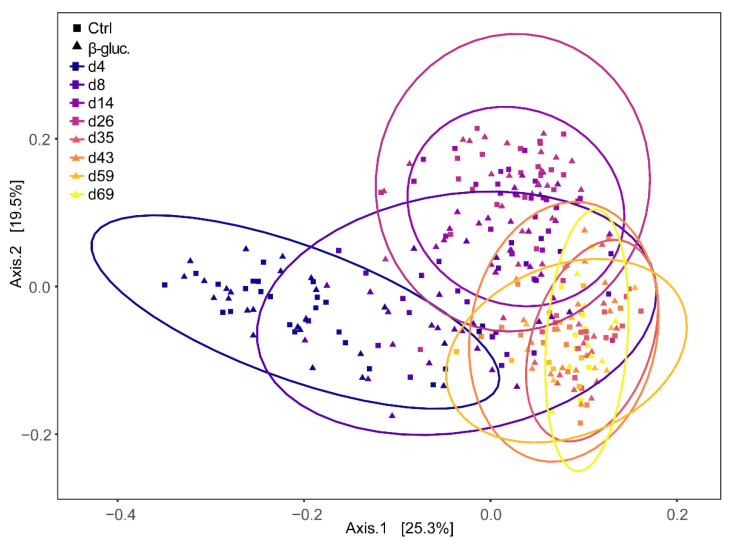
Principal Coordinates Analysis (PCoA) plot of beta-diversity based on weighted Unifrac dissimilarities in faecal samples over time (d4-69). Every dot represents a faecal sample from a single animal and samples are enveloped per time point. Control and β-glucan treated animals are represented by squares (■) and triangles (▲), respectively.

**Figure 5 microorganisms-08-01573-f005:**
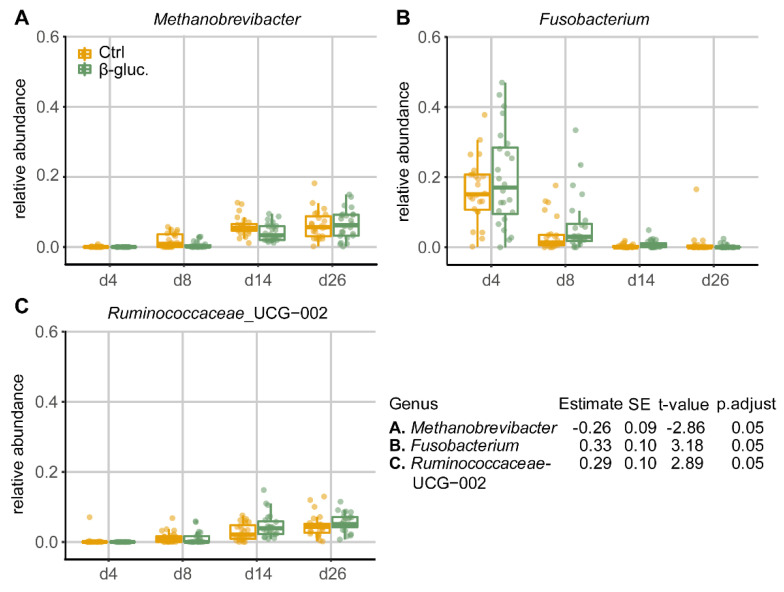
Relative abundances (by fraction) of differentially abundant genera in faeces during the pre-weaning period (d4-26). Shown genera resulted from comparing β-glucan animals to control animals using a GAMLSS model. Dietary intervention was used as the main variable for comparison and day of study and department were used as adjusting variables. Ear tag was used as an identifier and “fdr” was chosen as the method for multiple testing adjustment (*p*.adjust < 0.05).

**Figure 6 microorganisms-08-01573-f006:**
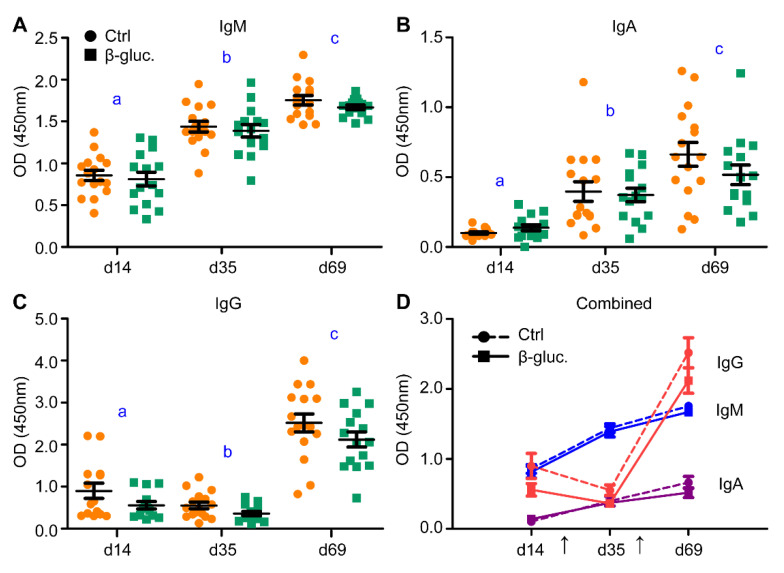
Levels of vaccine-specific IgM (**A**), IgA (**B**) and IgG (**C**) were measured in blood serum prior to vaccination (day 14), three weeks after vaccination (day 35) and 3 weeks after the booster vaccination (day 69). (**D**) incorporates the results of (**A**–**C**). The arrows in plot (**D**) indicate the time of vaccine (Salmoporc^®^) administration (d21 and d45). From every pen, two control animals (circles; ●, dashed lines) or two β-glucan treated animals (squares; ■, solid lines) were randomly selected and followed over time (*n* = 16 per treatment group). Every symbol represents a single observation. No significant differences were observed between the treatment groups. Letters (a, b, c) represent significant differences (*p* < 0.05) between time points.

**Figure 7 microorganisms-08-01573-f007:**
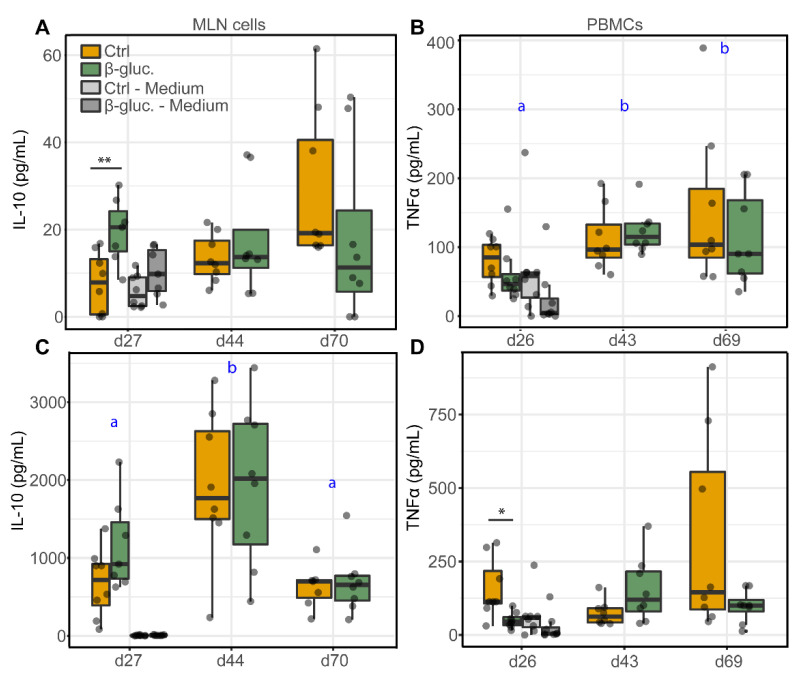
Levels of cytokines IL-10 (**A**,**C**) and TNFα (**B**,**D**) from stimulated MLN cells and PBMCs. MLN cells and PBMCs were stimulated with 10 µg/mL LPS (**A**,**B**) or 5 µg/mL Con-A (**C**,**D**), or left unstimulated (cell culture medium only) for 24 h. Significant differences between the treatment groups are indicated by asterisks (**; *p* < 0.01 and *; *p* < 0.05). Letters (a, b) represent significant differences (*p* < 0.05) between time points. Every dot represents a single animal from a different pen (*n* = 7 or 8 per treatment group) and error bars represent standard deviations. Statistical analysis was performed for every time point (T-test) and over time (Two-way ANOVA). Data were checked for normal distribution and equal variances and log-transformed when required.

**Figure 8 microorganisms-08-01573-f008:**
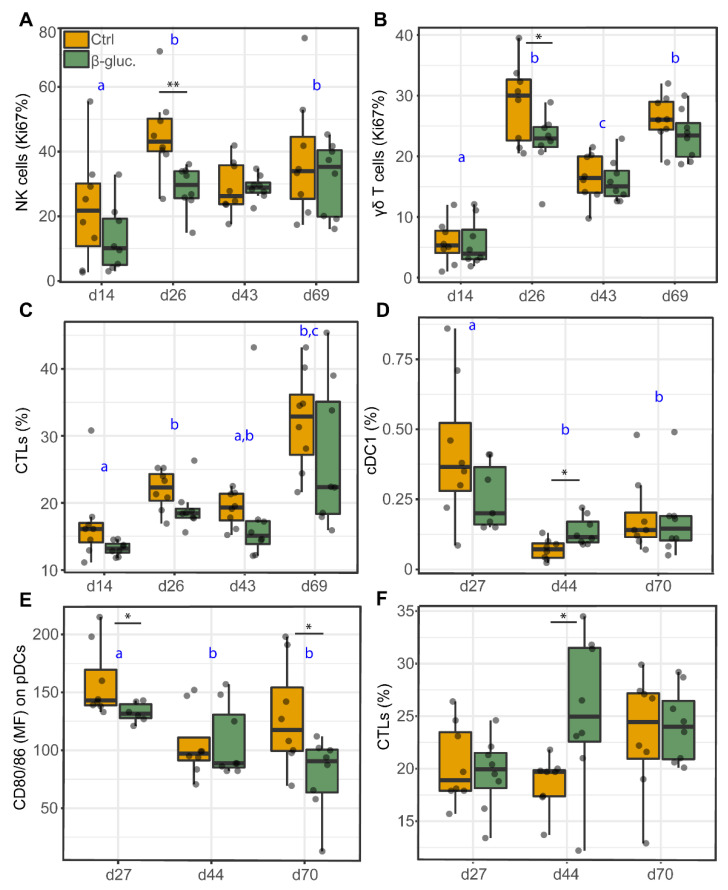
Cell analysis of different immune cell populations from PBMCs (**A**–**C**) and MLN cells (**D**–**F**). Significant differences between treatments are indicated by asterisks (**; *p* < 0.01 and *; *p* < 0.05). Letters (a, b, c) represent significant differences (*p* < 0.05) between time points. Data were presented as the percentage of proliferating cells (Ki67%; **A**,**B**), percentage of cells (%; **C**,**D**,**F**), or mean fluorescent intensity (MFI; **E**). Every dot represents a single animal from a different pen (*n* = 7 or 8 per treatment group) and error bars represent standard deviations. Statistical analysis was performed for every time point (T-test) and over time (Two-way ANOVA). All data were checked for normal distribution and equal variances and log-transformed when required.

**Table 1 microorganisms-08-01573-t001:** Antibodies used to identify DC subsets, NK cells and T cell subsets in PBMCs and mesenteric lymph node (MLN) cells.

Antibody	Host/Isotype	Clone	Fluorochrome	Company	Dilution
CD14	Mouse, IgG2b	MIL-2	FITC	Bio-Rad	1:50
CD172a	Mouse, IgG2b	74-22-15A	PE	BD Biosciences	1:40
CD4a	Mouse, IgG2b	74-12-4	PerCP-Cy5.5	BD Pharmingen^TM^	1:320
CADM1	Chicken, IgY	3E1	Biotin	MBL	1:200
Streptav.	n/a	n/a	BV421	BD Horizon^TM^	1:50
CD152 *	Mouse, IgG2a	n/a	APC	Ancell	1:320
CD3ε	Mouse, IgG2a	BB23-8E6-8C8	PE-Cy^TM^7	BD Pharmingen^TM^	1:160
CD4a	Mouse, IgG2b	74-12-4	PerCP-Cy5.5	BD Pharmingen^TM^	1:320
CD8a	Mouse, IgG2a	76-2-11	FITC	BD Pharmingen^TM^	1:10
FoxP3	Rat, IgG2a	FJK-16s	Alexa Fluor^®^ 700 **	eBioscience^TM^	1:20
Ki67	Mouse, IgG1	B56	BV421 **	BD Horizon^TM^	1:80
CD25	Mouse, IgG1	K231.3B2	Purified ***	Bio-Rad	1:200
γδ T cells	Rat, IgG2a	MAC320	PE	BD Pharmingen^TM^	1:20

*; Human CD152 (CTLA-4) murine Ig fusion protein was used to detect CD80 and CD86 on DC subsets. **; Antibodies against intracellular antigens. ***; The antibody CD25 (purified) was labelled with the ReadiLink™ Rapid iFluor™ 647 Antibody Labeling Kit (Aat Bioquest, Sunnyvale, CA, USA).

**Table 2 microorganisms-08-01573-t002:** Cell analysis of DCs, NK cells and T cells in PBMCs between treatment groups over time.

		Unpaired T-Test				Two-Way ANOVA
Subset	Phenotype	D14	D26	D43	D69	
		*p*-value [CI]	*p*-value [CI]	*p*-value [CI]	*p*-value [CI]	
pDCs (%)	CD14^−^CD4^+^CD172 a^+^CADM1^−^	ns	ns	ns	0.060 ^↓^ [−0.66 to 34.14]	T (<0.001), I (0.020)
pDCs (mat.)	CD14^−^CD4^+^CD172 a^+^CADM1^−^CTLA4-Ig^+^	ns	ns	ns	ns	T (<0.001)
cDC1 (%)	CD14^−^CD4^−^CD172 a^low^CADM1^+^	ns	ns	ns	ns	T (<0.001)
cDC1 (mat.)	CD14^−^CD4^−^CD172 a^low^CADM1^+^ CTLA4-Ig^+^	ns	ns	ns	ns	T (<0.001)
cDC2 (%)	CD14^−^CD4^−^CD172 a^+^CADM1^+^	ns	ns	ns	ns	T (0.092)
cDC2 (mat.)	CD14^−^CD4^−^CD172 a^+^CADM1^+^ CTLA4-Ig^+^	ns	ns	ns	ns	T (<0.001)
NK cells (%)	CD3^−^CD8α^+^	ns	0.060 ^↓^ [−0.41 to 15.68]	ns	ns	G (0.080), T (0.063)
NK cells (act.)	CD3^−^CD8α^+^CD25^+^	ns	ns	ns	ns	T (<0.001)
NK cells (Ki67%)	CD3^−^CD8α^+^Ki67^+^	ns	0.009 ^↓^ [5.15 to 28.53]	ns	ns	G (0.010), T (<0.001)
γδ T cells (%)	CD3^+^TCR-γδ^+^	ns	ns	ns	ns	T (0.005)
γδ T cells (Ki67%)	CD3^+^TCR-γδ^+^Ki67^+^	ns	0.049 ^↓^ [0.02 to 12.75]	ns	ns	G (0.034), T (<0.001)
CTLs (%)	CD3^+^TCR-γδ^−^CD8α^+^	ns	0.090 ^↓^ [−0.53 to 6.13]	ns	ns	G (0.008), T (<0.001)
CTLs (Ki67%)	CD3^+^TCR-γδ^−^CD8α^+^Ki67^+^	ns	ns	ns	ns	T (<0.001)
T helper (%)	CD3^+^TCR-γδ^−^CD4^+^	ns	ns	ns	ns	T (<0.001)
T helper (Ki67%)	CD3^+^TCR-γδ^−^CD4^+^Ki67^+^	ns	ns	ns	ns	T (<0.001)
Mem./Act. (%)	CD3^+^TCR-γδ^−^CD4^+^CD8α^+^	ns	ns	ns	ns	T (<0.001)
Mem./Act. (Ki67%)	CD3^+^TCR-γδ^−^CD4^+^CD8α^+^ki67^+^	ns	ns	ns	ns	T (<0.001)
Tregs (%)	CD3^+^TCR-γδ^−^CD4^+^CD25 ^high^Foxp3^+^	ns	ns	ns	ns	T (0.003)
Tregs (Ki67%)	CD3^+^ TCR-γδ^−^CD4^+^CD25 ^high^Foxp3^+^Ki67^+^	ns	ns	ns	ns	T (<0.001)

Ns; not significant, 0.05 < *p* < 0.1; trend, *p* < 0.05; significant. An unpaired Student’s T-test and a Two-way ANOVA were conducted to analyse the group effect per time point, and the effects over time, respectively. G; group effect, T; time effect, I; group-time interaction, act.; activation, mat.; maturation, Mem./Act.; Memory/Activated T cells. Arrows indicate if the values (e.g., percentage of cells) in the treatment group (β-glucan, *n* = 7 or 8) is significantly lower (^↓^) than the control group (*n* = 7 or 8). The 95% confidence intervals (CI) on the difference between the means are shown for all significant results or results which show a trend.

**Table 3 microorganisms-08-01573-t003:** Cell analysis of DCs, NK cells and T cells in MLN cells between treatment groups over time.

		Unpaired T-Test			Two-Way ANOVA
Subset	Phenotype	D27	D44	D70	
		*p*-value [CI]	*p*-value [CI]	*p*-value [CI]	
pDCs (%)	CD14^−^CD4^+^CD172 a^+^CADM1^−^	ns	ns	ns	T (<0.001)
pDCs (mat.)	CD14^−^CD4^+^CD172 a^+^CADM1^−^CTLA4-Ig^+^	0.034 [3.21 to 67.89] ^↓^	ns	0.024 [7.75 to 92.70] ^↓^	G (0.007), T (0.001), I (0.08)
cDC1 (%)	CD14^−^CD4^−^CD172 a^low^CADM1^+^	ns	0.011 [−0.11 to −0.02] ^↑^	ns	T (<0.001), I (0.028)
cDC1 (mat.)	CD14^−^CD4^−^CD172 a^low^CADM1^+^CTLA4-Ig^+^	ns	ns	ns	T (<0.001)
cDC2 (%)	CD14^−^CD4^−^CD172 a^+^CADM1^+^	ns	ns	ns	T (0.001)
cDC2 (mat.)	CD14^−^CD4^−^CD172 a^+^CADM1^+^CTLA4-Ig^+^	0.084 [−62.71 to 4.50] ^↑^	ns	ns	ns
NK cells (%)	CD3^−^CD8 a^+^	n/a	n/a	n/a	n/a
NK cells (act.)	CD3^−^CD8α^+^CD25^+^	n/a	n/a	n/a	n/a
NK cells (Ki67%)	CD3^−^CD8α^+^Ki67^+^	n/a	n/a	n/a	n/a
γδ T cells (%)	CD3^+^TCR-γδ^+^	ns	ns	ns	T (<0.001)
γδ T cells (Ki67%)	CD3^+^TCR-γδ^+^Ki67^+^	0.089 [−15.45 to 1.26] ^↑^	ns	ns	T (<0.001), I (0.086)
CTLs (%)	CD3^+^TCR-γδ^−^CD8α^+^	ns	0.035 [−13.03 to −0.60] ^↑^	ns	T (0.076)
CTLs (Ki67%)	CD3^+^TCR-γδ^−^CD8α^+^Ki67^+^	ns	ns	ns	T (<0.001)
T helper (%)	CD3^+^TCR-γδ^−^CD4^+^	ns	ns	ns	T (<0.001)
T helper (Ki67%)	CD3^+^TCR-γδ^−^CD4^+^Ki67^+^	0.079 [−12.53 to 0.77] ^↑^	ns	ns	T (<0.001), I (0.019)
Mem./Act. (%)	CD3^+^CD4^+^CD8α^+^	ns	ns	ns	T (<0.001)
	CD3^+^CD4 ^high^CD8α^low^	ns	ns	ns	T (0.002)
	CD3^+^CD4 ^high^ CD8α^low^ki67^+^	ns	ns	ns	T (<0.001)
	CD3^+^CD4 ^low^CD8α^high^	ns	ns	ns	ns
	CD3^+^CD4 ^low^ CD8α^high^ki67^+^	ns	ns	ns	T (<0.001)
Tregs (%)	CD3^+^TCR-γδ^−^CD4^+^CD25 ^high^Foxp3^+^	ns	ns	ns	T (<0.001)
Tregs (Ki67%)	CD3^+^ TCR-γδ^−^CD4^+^CD25 ^high^Foxp3^+^Ki67^+^	n/a	n/a	n/a	n/a

Ns; not significant, 0.05 < *p* < 0.1; trend, *p* < 0.05; significant. An unpaired Student’s T-test and a Two-way ANOVA were conducted to analyse the group effect per time point, and the effects over time, respectively. G; group effect, T; time effect, I; group-time interaction, act.; activation, mat.; maturation, Mem./Act.; Memory/Activated T cells. Arrows indicate if the values (e.g., percentage of cells) in the treatment group (β-glucan, *n* = 7 or 8) are significantly higher (^↑^) or lower (^↓^) than the control group (*n* = 7 or 8). The 95% confidence intervals (CI) on the difference between the means are shown for all significant results or results which show a trend.

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
