# Peer review of "Impact of Yeast-Derived β-Glucans on the Porcine Gut Microbiota and Immune System in Early Life"

_microorganisms, 2020, doi:10.3390/microorganisms8101573_

Round 1

Reviewer 1 Report

Review of the manuscript “Impact of yeast-derived β-glucans on the porcine gut 2 microbiota and immune system in early life” by Hugo de Vries submitted to “Microorganisms”.

The authors studied the effects of early life supplementation yeast beta-glucan on the gut microbiota and immune system of piglets as well as the dynamics of the microbiome and immune response in different time points. The topic is important due to the increasing evidences of early life diet impact  on the immune system function in a long lasting perspective. Beta-glucans are espetially imteresting bioactive compounds due to its strong immunomodulatory effect and the manuscript may be interesting for the readers of “Microorganisms”. Since the submitted manuscript is well written and organized, it is suitable for publication in “Microorganisms”.. However, there are some points to be considered before the publication in order to improve the manuscript. Please revise the submitted manuscript according to the comments listed below.

  • In the Introduction (line 72-77) the authors referenced the manuscripts that evaluated the effect of live yeast on the intestinal health. In the reviewer's opinion, these effects are completely different than observed after beta-glucan administration due to the possibility of gut colonization. I suggest excluding this information from the introduction
  • The study design is difficult to follow, I recommend to add (additional to the timeline of the experiment (Figure1)(it is also necessary)) schematic experimental design (with N of animal per group).
  • The animals received MacroGard as a beta-glucan source. The composition of MacroGard should be described in detail. According to section 2.3.1. MacroGard contains a minimum of 60% of beta-glucan. It is very important to compare the observed effect with the components present in the formulation. Based on the description the MacroGard is rather a yeast fiber with high beta-glucan content.
  • Another important issue is the beta-glucan characterization. As the authors mentioned in the Introduction the immunomodulatory (and probably also microbiome modulatory) effect of beta-glucan is dependent on its molecular weight. No information about the molecular weight of assessed beta-glucan has been provided. It should evaluate how the proportion of different molecular weight beta-glucan fraction is present in the MacroGard. The paragraph with beta-glucan characterization should be included in the Methods section.
  • How the authors chose the dose of beta-glucan supplementation? The selection of a dose of supplementation should be explained.
  • What route was beta-glucan administered? The authors wrote orally Was beta-glucan administered in a gastric tube or with the feed?
  • Whether the authors isolated bacterial DNA from fresh or frozen samples. How the authors avoided/got rid of host DNA.
  • Two-way ANOVA should be followed by post-hoc test. Which post-hoc test has been used by authors to detect differences between groups?
  • In the Figures that present the results of the study, the statistical significance should be included (in some figures it is so, in some it is not).
  • Figure 6 – wthat means what do the dashed lines mean?

Author Response

Dear editor,

We would like to thank you and the reviewers for the valuable comments to our manuscript “Impact of yeast-derived β-glucans on the porcine gut microbiota and immune system in early life” by Hugo de Vries 1,3,§, Mirelle Geervliet 2,§, Christine A. Jansen 4, Victor P.M.G. Rutten 4, Hubèrt van Hees 5, Natalie Groothuis 2, Jerry M. Wells 3, Huub F.J. Savelkoul 2, Edwin Tijhaar 2,§ and Hauke Smidt1,*,§

We have carefully considered the reviewers comments. Please find below a detailed point-by-point reply according to which we revised the manuscript. Please note that the line numbers given in this document correspond to the line numbers in the revised manuscript (which includes Track Changes).

Response to Reviewer 1 Comments:

Point 1: In the Introduction (line 72-77) the authors referenced the manuscripts that evaluated the effect of live yeast on the intestinal health. In the reviewer's opinion, these effects are completely different than observed after beta-glucan administration due to the possibility of gut colonization. I suggest excluding this information from the introduction.

Response 1: We agree with the reviewer’s remark. Consequently, the text regarding the use of live yeast has been removed (lines 73-75).

Point 2: The study design is difficult to follow, I recommend to add (additional to the timeline of the experiment (Figure1)(it is also necessary)) schematic experimental design (with N of animal per group).

Response 2: Thank you for this valuable suggestion. We made the following changes:

  • In addition to the timeline of the study we now included a schematic representation of the experimental design (Figure 1, panel B) with emphasis on the number of animals per treatment group over time. We changed the related text and figure caption accordingly (line 127-128 and lines 152-155).
  • We moved the sentence in which we refer to Figure 1 to help the reader (lines 127-128).
  • More details have been added to lines 180-181 regarding the number of faecal samples analysed per timepoint and in lines 184-185 the selection of faecal samples is better explained.
  • The Figure captions of Figures 6, 7 and 8 now include extra information regarding the number of animals selected per pen (Figure 6; lines 513 – 518, Figure 7; 546-547 and Figure 8; 572).

We are confident that these changes will help the reader to understand the study design.

Point 3: The animals received MacroGard as a beta-glucan source. The composition of MacroGard should be described in detail. According to section 2.3.1. MacroGard contains a minimum of 60% of beta-glucan. It is very important to compare the observed effect with the components present in the formulation. Based on the description the MacroGard is rather a yeast fiber with high beta-glucan content.

Response 3: Thank you very much for your suggestion. The product MacroGard is a commonly used product, sold as a health promoting feed additive, because of its high levels of purified β[1,3/1,6]-glucans. It indeed consists of 100% yeast (Saccharomyces cerevisiae) cell walls, containing a minimum of 60% β[1,3/1,6]-glucans. Other major components of MacroGard are lipids (max. 18%), proteins (max. 8%) and raw ash (max 10 %). These details have now been described in lines 161 – 166. In addition, in line 166-167 we mentioned that the batch did not contain lipopolysaccharides as was assessed in a previous study (reference included in manuscript). Immune modulating effects of MacroGard can therefore not be attributed to lipopolysaccharides.

Point 4: Another important issue is the beta-glucan characterization. As the authors mentioned in the Introduction the immunomodulatory (and probably also microbiome modulatory) effect of beta-glucan is dependent on its molecular weight. No information about the molecular weight of assessed beta-glucan has been provided. It should evaluate how the proportion of different molecular weight beta-glucan fraction is present in the MacroGard. The paragraph with beta-glucan characterization should be included in the Methods section.

Response 4: The molecular weight of beta-glucan is indeed one of the factors which could influence the effect on the immune parameters. The molecular weight of beta-glucan molecules in yeast (Saccharomyces cerevisiae), as present in MacroGard, is approximately 200 kDa. We now referred to this point in lines 162 and included a relevant reference. In addition, we also want to stress that the beta-glucans in MacroGard are not present as soluble beta-glucan molecules, but as part of fragment of the cell walls of yeast. That is, the beta-glucans in MacroGard are present in a particulate form. This is relevant because previous research demonstrated that especially particulate beta-glucans (as present in MacroGard) have shown to induce immunomodulatory effects.

Point 5. How the authors chose the dose of beta-glucan supplementation? The selection of a dose of supplementation should be explained.

Response 5: Thank you for attending to this point. We have now included the following text in the manuscript (lines 167-170): ‘The dosage of β-glucans was gradually increased every week with 50 mg, starting from 50 mg per administration in week one to 300 mg per administration in week six. These dosages of β-glucans were chosen by taking previous studies into consideration’. A reference to these studies is included in the manuscript (line 170). In addition, we consulted different animal nutrition experts and researchers from universities and institutions to further determine these dosages.

Point 6: What route was beta-glucan administered? The authors wrote orally Was beta-glucan administered in a gastric tube or with the feed?

Response 6: We agree that description of the administration route is not clear enough. We now explained in lines 159 and 160 that the piglets received the dietary interventions orally, by using disposable syringes. In addition, we also included the brand of the used syringes.

Point 7: Whether the authors isolated bacterial DNA from fresh or frozen samples. How the authors avoided/got rid of host DNA.

Response 7: DNA was isolated from frozen faeces and digesta, since all samples were stored at -80 degrees immediately after sample collection (line 183-184 and line 198-199). To further clarify this, the words ‘previously frozen (-80 °C)’ have been added to line 205. Amplification of host DNA was avoided by selecting primers that exclusively amplify the bacterial and archaeal 16S rRNA gene, as indicated in lines 216-219.

Point 8: Two-way ANOVA should be followed by post-hoc test. Which post-hoc test has been used by authors to detect differences between groups?

Response 8: Thank you for this comment. We used the Tukey post-hoc test. Since this was indeed not mentioned, we now included it in the materials & methods section (lines 350-352).

Point 9:  In the Figures that present the results of the study, the statistical significance should be included (in some figures it is so, in some it is not).

Response 9: Indeed, the statistics described in section 3.1.1. (Alpha Diversity) were not presented in Figure 2. We now included statistical significance between timepoints and between treatment groups in Figure 2. The horizontal line (Figure 2B) with corresponding asterisk represents the significant difference between treatment groups in the pre-weaning period, and the letters (Figure 2A and 2B) are used to indicate significant differences between time points. In addition, we added vertical lines to Figure 2 to separate the results of the faecal samples and the luminal samples (jejunum, ileum, caecum). We also added letters to Figure 6 to indicate significant changes of IgM, IgA and IgG over time. The Figure captions of both Figures 2 and 6 are adjusted accordingly (lines 428 – 430 and lines 517 – 518).

Point 10: Figure 6 – What do the dashed lines mean?

Response 10: Figure 6 (panel D) demonstrates levels of IgM, IgA and IgG over time. The dashed lines with a circle are representing the control animals. The solid line with the square are representing the B-glucans treated animals. To make this more clear we adjusted Figure 6 (panel D) accordingly, by adding a legend explaining the symbols.

Additional Comments from the authors:

During the review process we made some minor changes to the manuscript:

  • Throughout the document we corrected a few language errors we had missed in the original version.
  • One of the authors departments recently changed names and is corrected (line 10).
  • Minor changes were made to the abstract of the manuscript to improve its readability (lines 17 -32).
  • In section 2.4.4. (stimulation assay) I placed the concentrations of ConA and LPS later on in the text to avoid confusion about the final concentration used for this assay (lines 276-277 and line 280-281).
  • In section 2.4.5. (flow cytometry). We identified more clearly the use of beads and cells as controls for spectral overlap to avoid confusion by the reader (lines 300-301).
  • In table 2 and 3 we removed a small error (mat. is changed to act.). In addition, we explained the abbreviations used in the table caption (‘; activation, mat.; maturation, Mem./Act.; Memory/Activated T cells’) to avoid confusion by the reader.
  • We think that the first sentence of the discussion was too long and unclear. We now shortened the sentence to increase the readability at the start of the discussion (lines 587-590).
  • Letters were missing in the Figure caption of Figure 8 and are important to clarify which graphs refer to PBMCs and which graphs refer to MLN cells (lines 568-569).
  • The wording of lines 668-669 in the discussion section was not correct and is changed.
  • We used the abbreviation SCFAs, but we need to write the abbreviation in full when using it the first time. We now included the ‘Short chain fatty acids’ in line 609.
  • The colour scheme of figure 2 and figure 5 have been adapted so that the colours correspond to the other figures. This will help colour-blind people to read these figures.

Reviewer 2 Report

The study performed by Hugo de Vries, et al., aimed to study the impact of yeast-derived β-glucans on the porcine gut microbiota and the immune system in early life.”
The study design and data analysis are by the usual practice.

Overall, it is well conducted and well written.

I have some minor suggestions:
describe the different R-scripts you have coded and give details about the validation criteria for various statistical models.

Discuss the lack of power for some of your inferences.

Author Response

Dear editor,

We would like to thank you and the reviewers for the valuable comments to our manuscript “Impact of yeast-derived β-glucans on the porcine gut microbiota and immune system in early life” by Hugo de Vries 1,3,§, Mirelle Geervliet 2,§, Christine A. Jansen 4, Victor P.M.G. Rutten 4, Hubèrt van Hees 5, Natalie Groothuis 2, Jerry M. Wells 3, Huub F.J. Savelkoul 2, Edwin Tijhaar 2,§ and Hauke Smidt1,*,§

We have carefully considered the reviewers comments. Please find below a detailed point-by-point reply according to which we revised the manuscript. Please note that the line numbers given in this document correspond to the line numbers in the revised manuscript (which includes Track Changes).

Response to Reviewer 2 Comments:

Point 1: Describe the different R-scripts you have coded and give details about the validation criteria for various statistical models.

Response 1: This is a good suggestion. The following lines have been added to the manuscript (lines 312-315): ‘All R-scripts, data files and pdf files with extensive information on the performed analyses can be accessed through the following DOI: 10.4121/12999620. Alternatively, these files can be found under the following Github page: https://github.com/mibwurrepo/de-Vries-et-al-2020-porcine-study-beta-glucans’.

Point 2: Discuss the lack of power for some of your inferences.

Response 2: Thank you for this comment. The lack of power in relation to microbiota data from digesta is now described in the discussion section, see lines 622-624, and for immunological data this has been described in lines 691-692.

Additional Comments from the authors:

During the review process we made some minor changes to the manuscript:

  • Throughout the document we corrected a few language errors we had missed in the original version.
  • One of the authors departments recently changed names and is corrected (line 10).
  • Minor changes were made to the abstract of the manuscript to improve its readability (lines 17 -32).
  • In section 2.4.4. (stimulation assay) I placed the concentrations of ConA and LPS later on in the text to avoid confusion about the final concentration used for this assay (lines 276-277 and line 280-281).
  • In section 2.4.5. (flow cytometry). We identified more clearly the use of beads and cells as controls for spectral overlap to avoid confusion by the reader (lines 300-301).
  • In table 2 and 3 we removed a small error (mat. is changed to act.). In addition, we explained the abbreviations used in the table caption (‘; activation, mat.; maturation, Mem./Act.; Memory/Activated T cells’) to avoid confusion by the reader.
  • We think that the first sentence of the discussion was too long and unclear. We now shortened the sentence to increase the readability at the start of the discussion (lines 587-590).
  • Letters were missing in the Figure caption of Figure 8 and are important to clarify which graphs refer to PBMCs and which graphs refer to MLN cells (lines 568-569).
  • The wording of lines 668-669 in the discussion section was not correct and is changed.
  • We used the abbreviation SCFAs, but we need to write the abbreviation in full when using it the first time. We now included the ‘Short chain fatty acids’ in line 609.
  • The colour scheme of figure 2 and figure 5 have been adapted so that the colours correspond to the other figures. This will help colour-blind people to read these figures.